# Short-Term Effects of Atmospheric Pollution on Daily Mortality and Their Modification by Increased Temperatures Associated with a Climatic Change Scenario in Northern Mexico

**DOI:** 10.3390/ijerph17249219

**Published:** 2020-12-10

**Authors:** Rosa María Cerón Bretón, Julia Griselda Cerón Bretón, Jonathan W. D. Kahl, María de la Luz Espinosa Fuentes, Evangelina Ramírez Lara, Marcela Rangel Marrón, Reyna del Carmen Lara Severino, Martha Patricia Uc Chi

**Affiliations:** 1Chemistry Faculty, Autonomous University of Carmen, Campeche 24100, Mexico; jceronbreton@gmail.com (J.G.C.B.); mrangel@pampano.unacar.mx (M.R.M.); reyna.lara.sev@gmail.com (R.d.C.L.S.); muc@pampano.unacar.mx (M.P.U.C.); 2Mathematical Sciences, Atmospheric Science Group, University of Wisconsin-Milwaukee, Milwaukee, WI 53211, USA; kahl@uwm.edu; 3Atmospheric Sciences Center, National University of Mexico, Investigación Científica s/n, Ciudad de Mexico 04510, Mexico; marilu@atmosfera.unam.mx; 4Chemistry Faculty, Autonomous University of Nuevo Leon, Av. Universidad s/n, Ciudad Universitaria, San Nicolas de los Garza 66455, Mexico; evangelina.ramirez.lr@gmail.com

**Keywords:** relative risk index, mortality, criteria air pollutants, Monterrey, Mexico

## Abstract

Short-term effects of air pollution on the health of residents in the Metropolitan Area of Monterrey, Mexico were assessed from 2012–2015 using a time-series approach. Guadalupe had the highest mean concentrations for SO_2_, CO and O_3_; whereas Santa Catarina showed the highest NO_2_ concentrations. Escobedo and Garcia registered the highest levels for PM_10_. Only PM_10_ and O_3_ exceeded the maximum permissible values established in the Mexican official standards. Most of pollutants and municipalities showed a great number of associations between an increase of 10% in their current concentrations and mortality, especially for people >60 years. Different scenarios resulting from climatic change were built (increases of 5–25% in daily mean temperature), but only the increase of 25% (5 °C) showed a significant association with air pollutant concentrations and mortality. All pollutants and municipalities showed significant increases in relative risk indexes (RRI) resulting from an increase of 5 °C when people >60 years was considered. Results were comparable to those reported by other authors around the world. The RRI were low but significant, and thus are of public concern. This study demonstrated that the elderly is strongly threatened not only by atmospheric pollution but also by climatic change scenarios in warm and semiarid places.

## 1. Introduction

Air pollution is the main human health risk factor for respiratory and cardiovascular diseases [1]. High levels of atmospheric pollutants have both short- and long-term effects on human health, which can lead to an increase in morbidity and also in the number of deaths.

The effects of criteria air pollutants on human health are well known [1,2,3,4]. In recent years, epidemiological studies have documented an association between long and short-term exposure to air pollution and an increase in the risk of mortality and morbidity [1,5,6,7,8]. Besides the high levels of atmospheric pollutants, extreme temperatures could be a threat to public health and be related to mortality, especially among vulnerable groups. Devastating effects related to both heat waves and cold spells have been documented [9]. Future climatic change is considered one of the most serious threats to both human life and current lifestyle. Because of greenhouse-induced climate change, there is an increase in mean temperatures (for example, an increase in the magnitude and frequency of hot days) and their variability [10].

In the case of the association between daily mortality and temperature typically has a U-shape: mortality risk decreases from the lowest temperature to an inflection point, and then increases with higher temperatures [11]. The estimation of vulnerable population depends on population estimations in the future, their distribution, and their ability to adapt. Therefore, this effect could be different depending on weather patterns, latitude and pollution levels [12]. Unfortunately, studies in developing countries about the association between extreme temperatures, air pollution and mortality or morbidity are scarce.

Several authors have reported a significant association between heat and daily mortality, with some heterogeneity in their results, depending on local population characteristics and climate. Li et al. [12] and Chen et al. [13] reported that the effect of air pollutants on mortality is modified by temperature, with the highest effects found during the summer, especially for PM_10_, PM_2.5_ and O_3_. It has been reported the synergistic effects of heat and air pollution on morbidity [14,15]; whereas the effects of air temperature on climate-sensitive mortality and morbidity outcomes in the elderly. It has been demonstrated that there are population groups which are more susceptible to these effects, such as old people and people with pre-existing condition [16]. Yu et al. [17] reported a greater risk for all-causes heat-induced mortality (2–5% per 1 °C increase in temperature) in population >65 years compared to all-causes cold-induced mortality in populations >50 years (1–2% per 1 °C decrease in temperature). Despite its importance, in Mexico there is not enough information about this subject; therefore, this study aims to provide knowledge about the short-term effects of criteria air pollutants (CO, SO_2_, NO_2_, O_3_, and PM_10_) on daily mortality in the Metropolitan Area of Monterrey in Mexico, presenting a case study showing how these effects would be modified by increased temperatures during a climate change scenario.

## 2. Materials and Methods

### 2.1. Study Area

Although the Metropolitan Area of Monterrey (MAM) does not have an extreme climate, high temperatures are frequently recorded during the summer; therefore, heat waves (hot days) and the regular occurrence of air pollutant concentrations exceeding air quality standards could have synergistic effects on health population. According to Köppen and Geiger [18], the climate in Monterrey is classified as BSh, which corresponds to warm-semiarid. The average temperature is 22 °C with an annual mean precipitation of 604 mm, however, the highest temperatures occur between May and August reaching between 37 and 43 °C. In addition, during July and August, the phenomenon known as mid-summer drought commonly occurs, characterized by the lack of precipitation and constant heat. On the other hand, the lowest temperatures occur during December and January (between 8 and 9 °C), however, temperatures as low than −7.5 °C have been reported in sporadic events in which frost and snowfall occur. Monterrey is the third most populated Metropolitan Area in Mexico, constituted by 8 municipalities (See Figure 1): Cadereyta (1140.9 km^2^), Escobedo (148.9 km^2^), García (1032.1 km^2^), Guadalupe (117.7 km^2^), Monterrey (324.8 km^2^), Salinas Victoria (1334.2 km^2^), San Nicolás de los Garza (60.2 km^2^) and Santa Catarina (915.8 km^2^); with a total population of 4,437,646 inhabitants, with an urban and industrial development has resulted in an increase in air pollution [19].

According to the last emission inventory for Nuevo León State [19], the total emissions of NO_x_ were approximately 77,114 MG year^−1^, from which, 26.7% came from power plants, 9.06% from area sources, 47.47% from vehicular sources, and 16.77% from other mobile sources. In the case of SO_x_, total emissions reported were 99,901 MG year^−1^, from which 82.13% were released from power plants and petroleum refineries, 15.77% from area sources (use of residual fuel in the industrial and commercial sectors), 1.95% from vehicular sources, and 0.15% from other mobile sources. In addition, 137,352 MG year^−1^ of VOC’s were emitted to the atmosphere in the state of Nuevo León, which from, fixed sources contribute with 16.25%, area sources with 48.08% (domestic combustion of wood, and commercial and domestic use of solvents, and distribution of LP gas), vehicular sources with 34.65%, and other mobile sources with 1.02%. Total emissions for CO were 407,386 MG year^−1^, which from, 5.43% came from fixed sources, 5.77% from area sources, 87.16% from vehicular sources, and 1.64% from other mobile sources. In the case of PM_10_, 18,650 MG year^−1^ were released to the ambient air, with 57.11% being released from fixed sources (power plants and mineral industry), 25.94% from domestic combustion of wood and agricultural tillage, 8.77% from vehicular sources, and 8.18% from other mobile sources. At national scale, Nuevo León is ranked in fourth, fifth and sixth place as generator of emissions of VOC’s, NOx and CO, respectively.

### 2.2. Health Data

Epidemiological data on mortality were obtained from the National Health Information System (SINAIS) (www.dgis.salud.gob.mx). Selected variables were: year, month and day of occurrence, municipality of residence, cause of death, sex and age group of the deceased. Deaths occurring outside MAM were excluded. The study population was grouped by gender and by age group: <1 years, 1–4 years, 5–59 years, 60–74 years, and >75 years. Causes of death considered in this study were categorized according to the International Classification of Diseases (ICD10): (1) Respiratory diseases (from J00 to J99); (2) Cardiovascular diseases (from I00 to I99); (3) All causes: other causes besides of respiratory and cardiovascular diseases, excluding suicides, violent and accidental deaths.

### 2.3. Air Pollution Measurements

Daily measurements of SO_2_, CO, NO_2_, O_3_, and PM_10_ were obtained from the air quality monitoring network of Nuevo León (SIMA) from 1 January 2012 to 31 December 2015. SIMA performs a strict quality control to assure a good performance of the atmospheric monitoring network, considering factors such as zero variation and span, results of calibrations and adjustments made, operating and maintenance services history, unusual changes in climatic conditions, changes due to seasonal conditions, levels of other pollutants during the same period, and so on. Therefore, daily, errors as inaccuracies due to air quality sensors faults or mistakes in the human handling of the sensors are registered by the SIMA staff, so data are removed before the validation process. During the validation process, in a first stage, in the case of values that are statistically different from those expected values at a given time and location (errors and outliers) are also identified and removed. Therefore, to identify outliers, a temporal outlier detection method was applied. It compares various function curves at fixed time periods, so functional outliers detection is used to compare entire vectors of measurements (e.g., all observations in a month). In addition, a spatial outlier detection method was applied, in which an observation is compared to the observations in its spatial neighborhood. In a second stage, all data were subjected to analysis, excluding data from monitoring stations with less than 75% of complete data for at least one pollutant during the study period. Missing values were generated using the MCMC (Markov Chain Monte Carlo) multiple imputation method and the NIPALS (Nonlinear estimation by Iterative Partial Least Square) approach using XLSTAT (https://www.xlstat.com/en/) (Details about imputation procedure are included in supplemental materials). Air quality was assessed by comparison with current national standards, and the Friedman test was used to determine if there were significant differences in mean concentrations for criteria air pollutants among different municipalities.

### 2.4. Confounding Variables Considered in the Study Cases

Temperature and relative humidity were used as confounding variables considering two study cases. The first, to assess the effect of a future increase of air pollution in MAM on daily mortality, with scenarios in which daily concentrations for criteria air pollutants were increased by 10%. The second, a climatic change scenario in which the response to atmospheric pollution was modified by high temperatures considering a susceptible population sub-group (over 60 years).

### 2.5. Estimation of the Association between Atmospheric Pollution and Daily Mortality

The methodology used in this work is described by Cerón et al. [8]. To carry out the statistical analysis, the following variables were considered:
(a)Response variables: Number of deaths registered for each municipality during the study period.(b)Explanatory variables: Criteria Air Pollutants (daily average concentrations of SO_2_, CO, NO_2_, O_3_ and PM_10_ for each municipality during the study period) and Meteorological variables (daily average values of temperature and relative humidity for each municipality during the study period).(c)Control variables: This was carried out by introducing seasonality indicator variables considering two strata: cold months and warm months.(d)Confounding variables: Average daily values of relative humidity and temperature.

A Principal Component Analysis (PCA) was applied to the daily average data series of mortality, criteria pollutants and meteorological variables in order to study the associations among variables. From biplot graphs, the principal components contributing to the highest percentage of variability of the data were identified and daily mortality time series were smoothed. To explain the fluctuations of daily number of deaths with respect to explanatory and confounding variables, a Poisson regression model was constructed according to methodology described by APHEA(Air Pollution and Health: A European Approach) and EMECAM (Spanish Multicenter Study on Relationship between Atmospheric Pollution and Mortality) projects [7,9,20], applying a Multiple Linear Regression (MLR), self-correlation functions between residuals and cross-correlation to all study variables (for each cause of death and for each pollutant) [21]. From the basal model, an equation was obtained to determine the magnitude of the association between daily mortality and the variation of the average daily levels of atmospheric pollution, by using the same methodology reported by Cerón et al. [8]. The selected variables to be included in the basal model were those that contributed in a significant way to the variability of the dependent variable (daily mortality). This procedure was applied considering all causes, age group and specific cause of death. The manifestation of the effects of atmospheric pollution on daily mortality is not immediate, it is necessary to consider lag periods (presumably short). To decide the lag term in the daily mean values for temperature and relative humidity, it was necessary to perform an analysis of cross correlations for data series: daily mortality vs. daily mean temperature, and daily mortality vs. daily mean relative humidity. Later, the most statistically significant lags were selected (if the regression coefficient showed *p* < 0.10). In this way, when such variables were introduced to the Poisson model, Pearson residuals were reduced (residual autocorrelation), and it was manifested in the function of simple autocorrelation. Therefore, meteorological variables were lagged by up to 7 days selecting those more significant. In the cases in which the independent variables significantly influenced the response variable (for a 95% confidence interval and *p* < 0.05), this effect was evaluated by the beta coefficient (β) of each independent variable. From the base model, the β coefficient values were obtained and used to estimate the relative risk indexes (RRI) as reported by Cerón et al. [8].

The Poisson model which relates the response variable to different independent variables is:(1)$$\ln(E_{y\;})\;=\;\beta_{o\;}+\;\overset{n}{\underset{i=1}{\sum }}\beta_{i}\;X_{ij}$$
where, *Ey* is expected number of daily deaths, *β_o_*, *β_i_* are constants of the model and *X_ij_* are the explanatory variables. From the base model, the *β* coefficient values are obtained and used to estimate the relative risk indexes using the following equation:(2)$$RRI_{i}\;=\;e^{\beta i}$$
where, *RRI* is the relative risk index associated to the explanatory variable *i* per unit of increase of this variable, and *β_i_* is the regression coefficient associated with the explanatory variable *i* in the model.

The next step was to generate different scenarios in which atmospheric pollutants concentrations increased by 10% (one-at-a-time and keeping the remaining variables unchanged), obtaining *β* and *RRI* values for daily mortality attributed to respiratory, circulatory, and all causes (>60 years). In addition, to assess the effect of modification induced by heat because of climatic change, additional scenarios were generated, considering increases of 5, 10, 15, 20, and 25% in daily mean temperatures. Again, the remaining variables were kept unchanged, and β and RRI values for mortality were obtained considering respiratory, circulatory, and all causes (>60 years) for mortality. Both procedures were performed for each pollutant and for each municipality considered in the study.

## 3. Results

### 3.1. Health Impact Assessment

#### 3.1.1. Health Database

The municipalities that showed the higher relative contribution to the total daily mortality during the study period were Monterrey, Guadalupe and San Nicolás de los Garza (Figure 2a). Considering all causes and gender, in all municipalities excepting Escobedo, Cadereyta and García, women >75 years showed the highest number of deaths in comparison with males (Figure 2b). A higher mortality by respiratory causes in people >75 years was observed, being higher in men (Figure 2c). Finally, mortality by circulatory causes in people >75 years was higher in women (Figure 2d).

#### 3.1.2. Exposure Database

Air quality was assessed for each municipality by comparing the daily maximum concentrations for each criteria air pollutant with the latest updates of the maximum permissible values established in Air Quality Mexican Standards (AQMS) (Figure 3).

Considering the whole period, Santa Catarina was the municipality with the highest mean concentration for SO_2_ (7.25 ppb) whereas Guadalupe showed the highest maximum value. In the case of CO, Guadalupe was the municipality with the highest mean concentration (1.16 ppm), whereas Guadalupe and San Nicolás de los Garza showed the highest maximum values. Santa Catarina showed the highest mean concentration for NO_2_ (18.27 ppb) and the highest maximum value considering the whole period. Regarding to O_3_, García was the municipality that registered the highest mean value (27.22 ppb), whereas Guadalupe and Escobedo showed the highest maximum values. Finally, the highest concentration for PM_10_ was registered in García (88.53 µg/m^3^), being García and Escobedo the municipalities that showed the highest maximum values.

All municipalities in MAM exceeded the reference value in the Air Quality Mexican Standards for O_3_ and PM_10_ (Figure 3).

### 3.2. Association between Atmospheric Pollutants and Daily Mortality

#### 3.2.1. Multivariate Analysis and Multiple Linear Regression

From the PCA analysis considering all causes, daily mortality showed significant factors for García with PM_10_ and temperature. In the case of Guadalupe, O_3_ and temperature showed a significant association with daily deaths. On the other hand, Monterrey had factors with highest loads for daily mortality with CO, humidity and temperature. In the case of San Nicolás de los Garza, NO_2_, CO and humidity were significantly associated with daily mortality, whereas in the case of Santa Catarina, CO and humidity showed a significant association with daily mortality. From the MLR analysis for all causes, it was found that Guadalupe (R^2^ = 0.5225) and Monterrey (R^2^ = 0.6810) showed the highest coefficients, suggesting that in these municipalities, explanatory variables (SO_2_, CO, NO_2_, O_3_, PM_10_,) explained 52.25% and 68.10%, respectively, of the variability of daily mortality. From the bivariate analysis, it was found that in the case of SO_2_, only Santa Catarina showed a significant Pearson correlation coefficient (0.3431) with daily mortality. From analysis of sum of squares type III (MLR SC Type III), it was found that Escobedo, Guadalupe and Santa Catarina showed values lower than statistical test (F < 0.0001), therefore, it can be concluded that SO_2_ provides significant information to the prediction model for daily mortality in these municipalities.

For CO, it was found that Guadalupe and Monterrey showed significant values of Pearson coefficients with daily mortality, 0.5120 and 0.5209, respectively. From MLR and analysis of sum of squares type III, it was found that Escobedo, Guadalupe, Monterrey and Santa Catarina, showed values lower than the statistical test (F < 0.0001), concluding that CO provides significant information to the prediction model for daily mortality in these municipalities. In the case of NO_2_, only San Nicolás de los Garza showed significant values of Pearson correlation coefficient (0.4149) with daily mortality. From MLR analysis, it was found that Escobedo, Monterrey, San Nicolás de los Garza and Santa Catarina showed values lower than statistical test, suggesting that NO_2_ provides enough information to explain the variability of daily mortality in these municipalities. In the case of ozone, low correlations (Pearson coefficient) with daily mortality were found for Escobedo, Guadalupe. In the case of PM_10_, it was found that no municipality in MAM showed significant values for Pearson coefficients with daily mortality. From the Fisher tests and the analysis of sum of squares, it was observed that, only Guadalupe showed values lower than the statistical tests, concluding that PM_10_ provide significant information to explain daily mortality in this municipality. For humidity, only Escobedo presented the highest Pearson correlation (0.4718) in the bivariate analysis, whereas, from the MLR analysis, it was found that García, Escobedo, Monterrey and Santa Catarina showed values lower than the statistical test, concluding that humidity accounts significantly to daily mortality in these municipalities. Finally, in the case of temperature, inverse significant correlations were obtained for García (−0.4169), Escobedo (−0.4307), Guadalupe (−0.6249), Monterrey (−0.7550), San Nicolás de los Garza (−0.5154) and Santa Catarina (−0.4054). The MLR analysis demonstrated that temperature accounts significantly in daily mortality in these municipalities.

#### 3.2.2. Estimation of Relative Risk Index (RRI)

In Figure 4, Figure 5, Figure 6, Figure 7 and Figure 8, the number of deaths, and the percentage of risk of daily mortality by all causes as a result of an increase of 10% in mean daily concentrations are showed for each municipality in MAM during 2012–2015. SO_2_: Considering all causes, the highest upper limit values (UL) of RRI were obtained for people >60 years for Cadereyta, Escobedo, Guadalupe and San Nicolás. When specific causes of daily mortality were considered, the highest UL values were found for Cadereyta (respiratory causes) and Escobedo (circulatory causes) (Figure 4). CO: A greater number of associations with daily mortality was found for elderly, being higher for Escobedo. Cadereyta and Escobedo showed the highest UL values for the specific causes of respiratory and circulatory diseases, respectively (Figure 5). NO_2_: Considering all causes, RRI values (UL) were higher for people from 0 to 59 years in Cadereyta, and for people >60 years in San Nicolás de Los Garza, the latter being very high. Considering specific causes of death, UL values were higher for respiratory causes in Escobedo and Monterrey, and for circulatory causes in Escobedo and San Nicolás de los Garza (Figure 6). O_3_: UL values for RRI considering mortality by all causes were higher for age group 0–59 years in García and for people >60 years in Escobedo. For daily mortality by respiratory causes, only Monterrey showed a significant association, whereas for circulatory causes San Nicolás de los Garza had the highest UL value (Figure 7). PM_10_: Escobedo showed significant associations considering all causes of death (for the age groups 0–59 years and >60 years) and for circulatory causes. San Nicolás de los Garza showed the highest UL value for RRI for respiratory causes (Figure 8).

From the analysis by age group, most pollutants and municipalities considered in this study showed a great number of associations with a significant magnitude for people older than 60 years (subgroups; 60–74 years and >75 years), suggesting that this group of population is more vulnerable to the effects of atmospheric pollution.

### 3.3. Modification of the Effects by Extreme Temperatures

#### Estimation of Relative Risk Index (RRI)

Regarding temperature, when the age group of people >60 years was considered, higher relative risk indexes due to respiratory diseases were found. With an increase of 10% in daily mean temperature, relative risk indexes were higher for the elderly, when mortality by all causes was considered during warm months. From this first approach, different scenarios were built to test the effect of increases in daily mean temperature of 5, 10, 15, 20 and 25%. RRI values found did not show significant differences when mortality by all causes and all age groups were considered, except the age group of people >60 years with a 25% increase in daily mean temperature (approximately 5 °C). For this reason, the aim of this section was to assess the synergic effect because of an increase in daily mean temperature of 5 °C on daily mortality by all causes, for both, all age groups and the subgroup of population >60 years.

SO_2_: For all age groups, significant increases in RRI values were found with the 25% increase in daily mean temperature for Escobedo (0.98%), Guadalupe (0.19%), San Nicolás (0.30%) and Santa Catarina (0.89%). For people > 60 years, the increases in RRI related to extreme temperatures were significant in Cadereyta (0.18%), Escobedo (6.02%), Guadalupe (0.20%), and San Nicolás (0.34%). CO: For all causes and all population subgroups, it was observed a very slight increase in Escobedo (0.004%), Guadalupe (0.002%), and Salinas Victoria (0.006%); even with some decreases for Monterrey (0.003%), San Nicolás de los Garza (0.002%) and Santa Catarina (0.0017%). In the case of people >60 years, increases in RRI values were obtained for Escobedo (0.01%), Guadalupe (0.01%), Salinas Victoria (0.01%) and Santa Catarina (0.042%), while in Monterrey and San Nicolás, no change was observed. NO_2_: RRI values for all causes of death and all age groups showed increases for Escobedo (0.13%), Monterrey (0.07%), Salinas Victoria (0.01%) and San Nicolás de los Garza (0.02%). Relative risk indexes increased in Escobedo (0.21%), Monterrey (0.08%), Salinas Victoria (0.42%), San Nicolás (0.20%) and Santa Catarina (0.08%) when the age group >60 years was considered; showing significant increases in comparison when all causes of death and all age groups were considered. O_3_: it was observed a moderate increase, being higher in Escobedo (0.20%), Guadalupe (0.09%), and Santa Catarina (0.15%) for all causes of death and all age groups. The increases in RRI values in people >60 years were higher than those found for all age groups in Escobedo (0.24%), Guadalupe (0.11%), Monterrey (0.08%) and Santa Catarina (0.17%). PM_10_: Relatively low increases were obtained for all causes, being highest in Escobedo (0.07% for all ages, and 0.1% for people >60 years).

## 4. Discussion

### Comparison with Other Epidemiological Studies

SO_2_: From Figure 9, it can be observed that Cadereyta (by respiratory causes) and Escobedo (by circulatory causes) presented the highest RRI values in MAM for SO_2_. Excepting Cadereyta and Escobedo, the majority of the municipalities in MAM showed RRI values and 95% IC comparable to those reported in Lyon, France [22]; in thirteen Spanish cities [23], in Zaragoza, Spain [24], in Sao Paulo, Brazil [25], in North Korea [26] and in Chongqing, China [27]. In the case of Cadereyta and Escobedo, the magnitude of the association between an increase of 10% in SO_2_ on daily mortality was higher than those reported by other authors and only comparable to that reported in Barcelona, Spain [28].

CO: RRI and 95% IC values estimated for municipalities of MAM were significantly lower than those reported by other authors, suggesting that the current relative risk associated to an increase of 10% in CO levels is not a problem in MAM (Figure 10). This behavior could be explained from the heterogeneity found among cities due to the specific characteristics of each city. Additionally, some studies have reported no associations or not significant associations between CO and cardiovascular diseases, perhaps because of the limited sizes of the study populations and different subtypes of cardiovascular diseases considered [29,30,31].

NO_2_: The magnitude of the association between daily mortality and an increase of 10% in NO_2_ concentrations (Figure 11) was higher in Escobedo (for >60 years, circulatory and respiratory causes), Salinas Victoria and San Nicolás de los Garza for elderly. RRI’s values were lower than those found in Sao Paulo [25], North Korea [26], Barcelona [28], Canadian cities [32], and Panama City [33]. In the case of people >60 years, RRI and 95% IC values were comparable to those reported thirteen Spanish cities [23], European cities [34,35], and 272 cities in China [36].

O_3_: From Figure 12, it can be observed that RRI values as a result of an increase of 10% in O_3_ levels were higher in Escobedo, Guadalupe and Santa Catarina (for >60 years) and in Monterrey (by respiratory causes). In general, the magnitude of the association between daily mortality and O_3_ levels was significantly lower than those reported in Sao Paulo [25], Barcelona [28], Canadian cities [32], Panama City [33] and Hong Kong and Taipei [37]. On the other hand, for the age group >60 years, the RRI and 95% IC values were comparable to that reported in Stockholm [38], and eight cities in USA [39]. Finally, when respiratory and circulatory causes were considered, RRI values in MAM were lower than those found in some cities in USA [40].

PM_10_: Escobedo (for >60 years, and circulatory causes) and San Nicolás (by respiratory causes) showed the highest risk values in daily mortality in MAM associated with an increase of 10% in PM_10_ levels (Figure 13). These values were significantly lower than RRI values reported for Panama City [33], and lower than those reported in in Sao Paulo [25], in North Korea [26], in Chongqing, China [28], European cities [34], in Canadian cities [32], and in Temuco and Pudahuel, Chile [41]. When elderly people were considered, it was found that RRI values in MAM were comparable to those reported in Stockholm [38], in European cities [42], and in Tehran [43].

Basu et al. [11] have used identical methods to assess the effect estimation through different US cities even with different ranges of apparent temperature. This threshold value (in this case, heat threshold) is the value of apparent temperature which corresponds to a change in the effect estimation; unfortunately, in this study, heat threshold for MAM was not estimated. However, McMichael et al. [44] reported a heat threshold for Monterrey City of 31 °C (95% IC 31–33 °C), reporting an increase of 18.8% (95% IC 13–25%) in mortality for each °C above heat threshold. In this study, the risk associated with an increase of 5 °C in daily mean temperature in MAM was higher in Escobedo (UL 1.49%), Guadalupe (UL 1.15%), San Nicolás (UL 1.24%) and Santa Catarina (UL 2.83%); being these increases lower than reported by McMichael [44] in 2008 for Monterrey City. However, in all municipalities in MAM, except Santa Catarina (which was higher), this association was comparable to those reported in Mexico City and Santiago City [44], in 9 cities in USA [45], and in different regions in USA [46], in which an increase of 5 °C was considered.

## 5. Conclusions

Since these municipalities are the two largest in MAM Daily mortality was higher in Monterrey and San Nicolás. Daily mortality by all causes was higher in women than in men. Considering respiratory causes, daily mortality was higher in men >75 years; whereas for circulatory causes, mortality was higher for women >75 years.

The statistical analysis of daily mortality in MAM during 2012–2015 showed an increase in number of deaths when criteria pollutants were increased in 10% on the same day and on the seven previous days. Considering all causes, Escobedo, Santa Catarina, Guadalupe and Salinas Victoria were the municipalities in MAM which exhibited the highest values for RRI for most of the pollutants. All pollutants and municipalities showed associations with a significant magnitude for people older than 60 years. Taking into account the specific age group of >60 years, Escobedo showed the highest RRI and 95% IC values for all criteria air pollutants.

On the other hand, as a result of an increase of 10% in the daily mean temperature, RRI values were higher for elderly when mortality by all causes was considered. However, when different scenarios with increases in daily mean temperature were considered, only an increase of 25% exhibited significant differences in comparison with the remaining scenarios. When the assessment considered only the age group of >60 years, the obtained increase was very high and significant for SO_2_ in Cadereyta, Escobedo, Guadalupe and San Nicolás; whereas for CO, higher increases were obtained in Escobedo, Guadalupe and Salinas Victoria. In addition, significant increases in RRI values were found in the most of municipalities for NO_2_; and in Escobedo, Guadalupe, Santa Catarina and Monterrey for O_3_. Finally, daily mortality associated with PM10 was sensitive to an increase of 5 °C in daily mean temperature in Escobedo, Guadalupe and Salinas Victoria. In the case of increases of 5 °C in daily mean temperature, RRI values were lower than those reported in other studies for increases of 1 °C, but comparable to Mexico City, Santiago City and some cities and regions in USA for increases of 5 °C.

This research provides a base line to control the current air pollution levels in MAM, especially with respect to SO_2_, O_3_ and PM_10_; considering the large size of population exposed, even when observed associations were small but significant, RRI values found are of public concern. This study demonstrated that people >60 years are within a population group that is strongly threatened not only by atmospheric pollution but also by scenarios that could result from climatic change. Even though temperatures in Mexico are temperate and extremes values are not common (unlike mid-latitude sites), the expected increase in daily mean temperature as a result of climatic change could result in a higher number of deaths, especially in the population older than 60 years. The implementation of protective strategies could have a positive effect on population health, therefore, the measures necessary to prevent the harmful effects of temperature fluctuations should consider those specifically aimed at vulnerable groups (as homes with special design, and meteorological warning systems). It has been demonstrated that populations in warmer/colder regions are more resilient to the effects of heat/cold, suggesting that there is some potential for acclimatization. From this, it could be concluded that the effects in Mexico won’t be so drastic as expected in other latitudes; however, it is well known that patterns of temperature-related mortality are influenced not only by climate factors but also non-climate factors as socioeconomic (lack of air conditioner, access to health services, quality of public health systems, and so on), therefore, populations in developing countries could be probably more vulnerable to the direct impact of climate change and extremes of temperature in the future.

## Figures and Tables

**Figure 1 ijerph-17-09219-f001:**
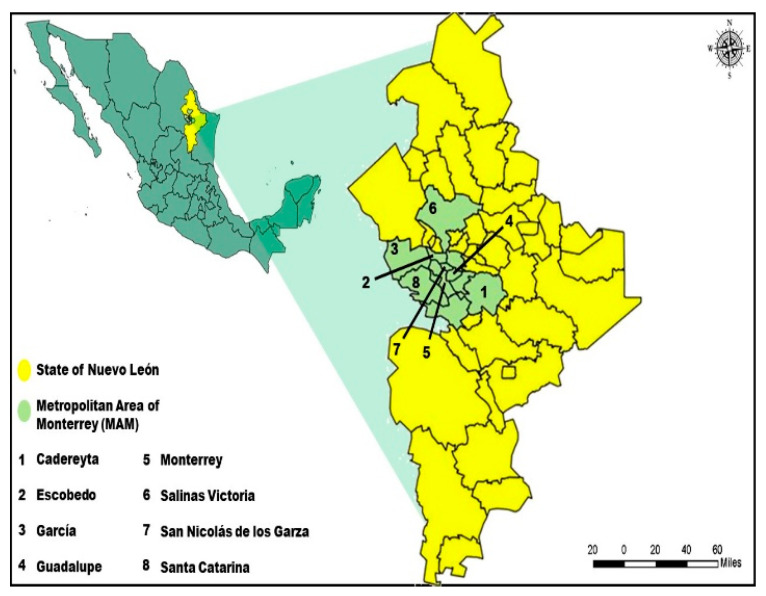
Location of municipalities in MAM.

**Figure 2 ijerph-17-09219-f002:**
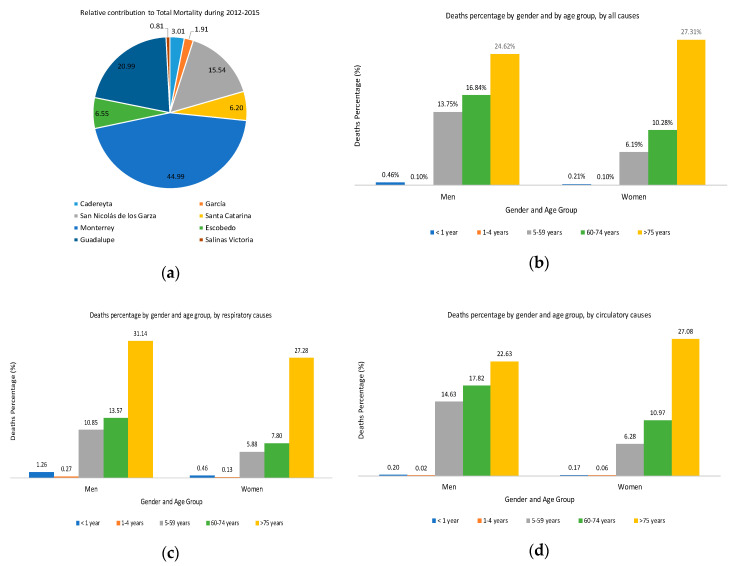
(**a**) Relative contribution to total Mortality, (**b**) Deaths percentage by gender and by age group, by all causes, (**c**) Deaths percentage by gender and by age group, by respiratory causes, (**d**) Deaths percentage by gender and by age group, by circulatory causes.

**Figure 3 ijerph-17-09219-f003:**
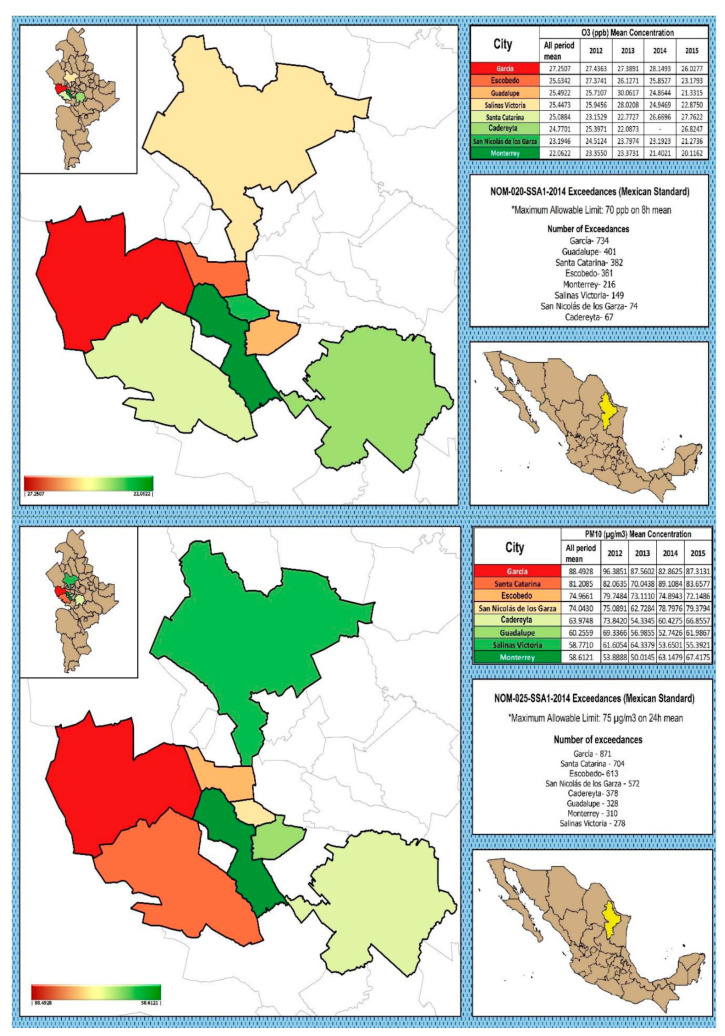
Map of mean concentrations for O_3_ and PM_10_ showing the exceedances.

**Figure 4 ijerph-17-09219-f004:**
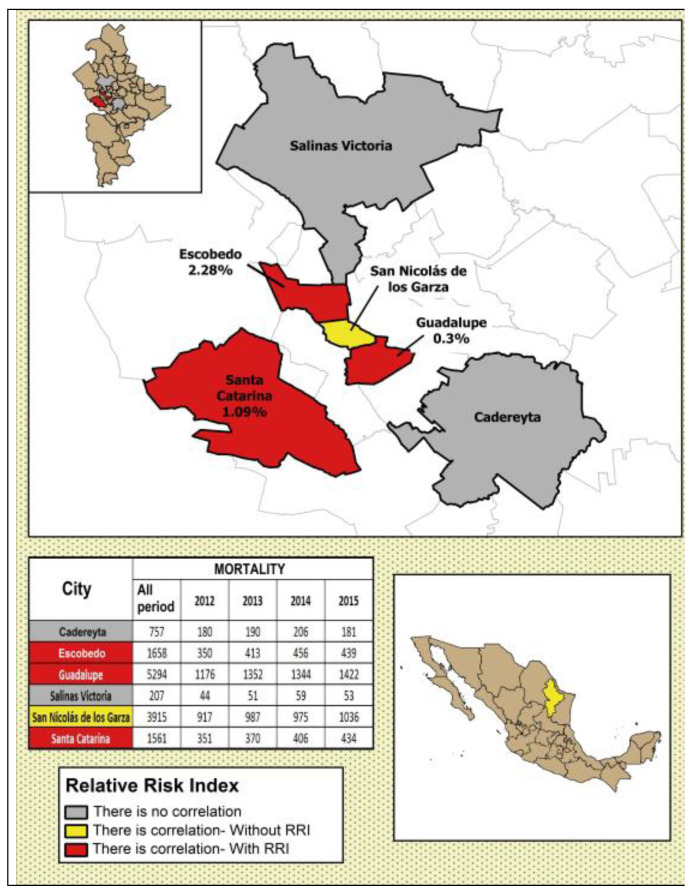
Number of deaths and the percentage of risk of daily mortality by all causes as a result of an increase of 10% in mean daily concentration for SO_2_.

**Figure 5 ijerph-17-09219-f005:**
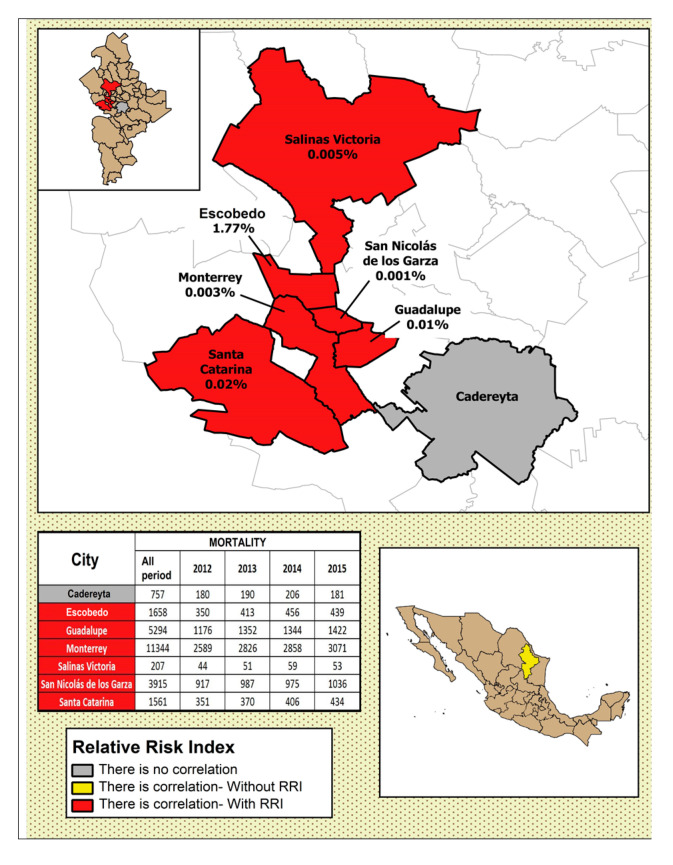
Number of deaths and the percentage of risk of daily mortality by all causes as a result of an increase of 10% in mean daily concentration for CO.

**Figure 6 ijerph-17-09219-f006:**
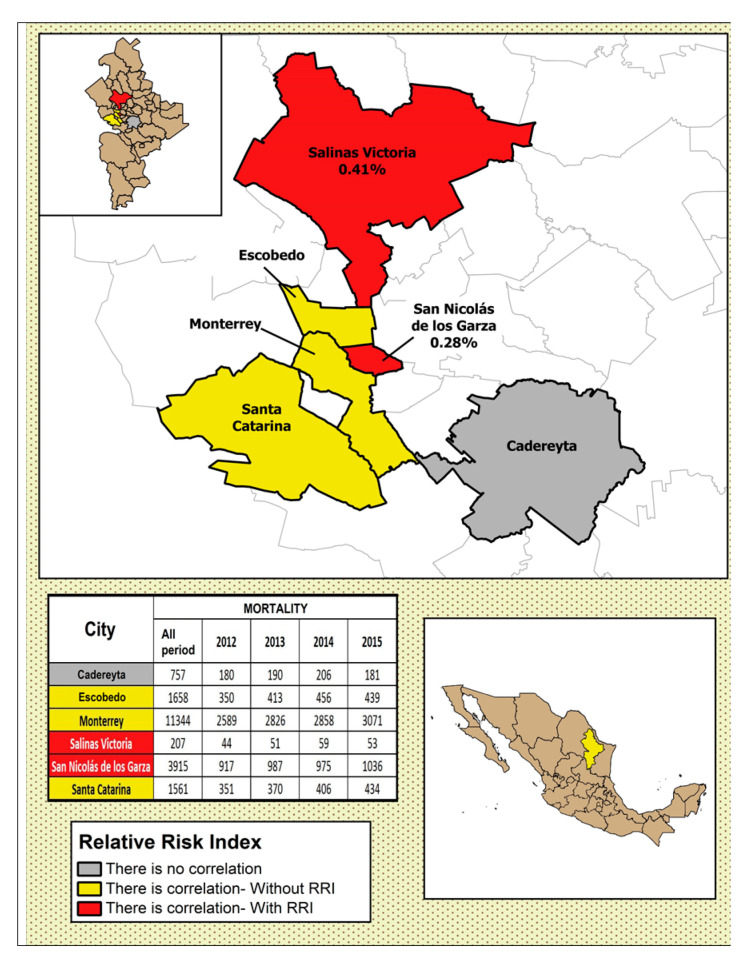
Number of deaths and the percentage of risk of daily mortality by all causes as a result of an increase of 10% in mean daily concentration for NO_2_.

**Figure 7 ijerph-17-09219-f007:**
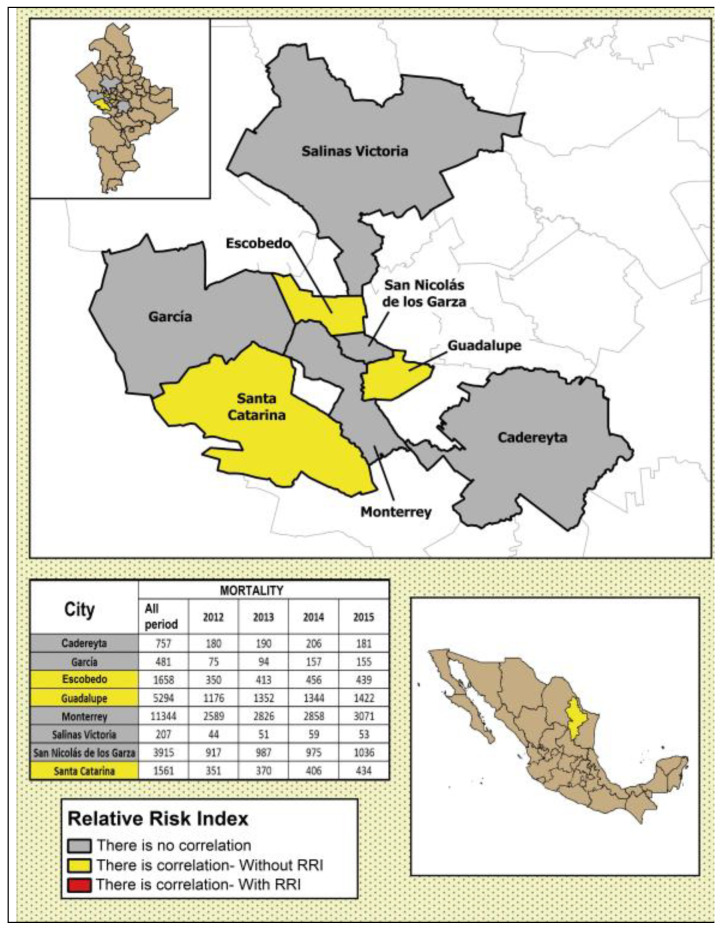
Number of deaths and the percentage of risk of daily mortality by all causes as a result of an increase of 10% in mean daily concentration for O_3_.

**Figure 8 ijerph-17-09219-f008:**
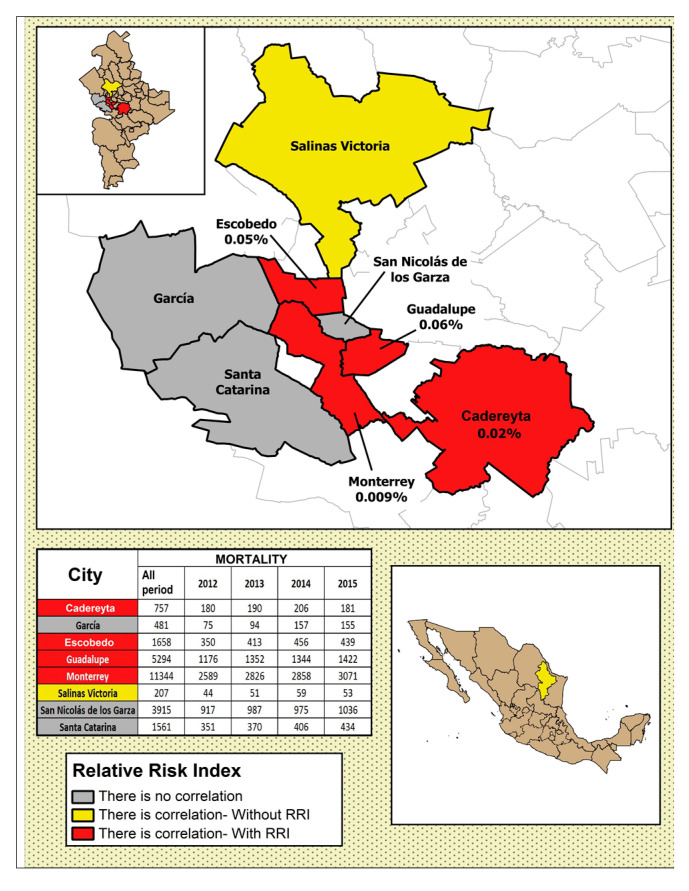
Number of deaths and the percentage of risk of daily mortality by all causes as a result of an increase of 10% in mean daily concentration for PM_10_.

**Figure 9 ijerph-17-09219-f009:**
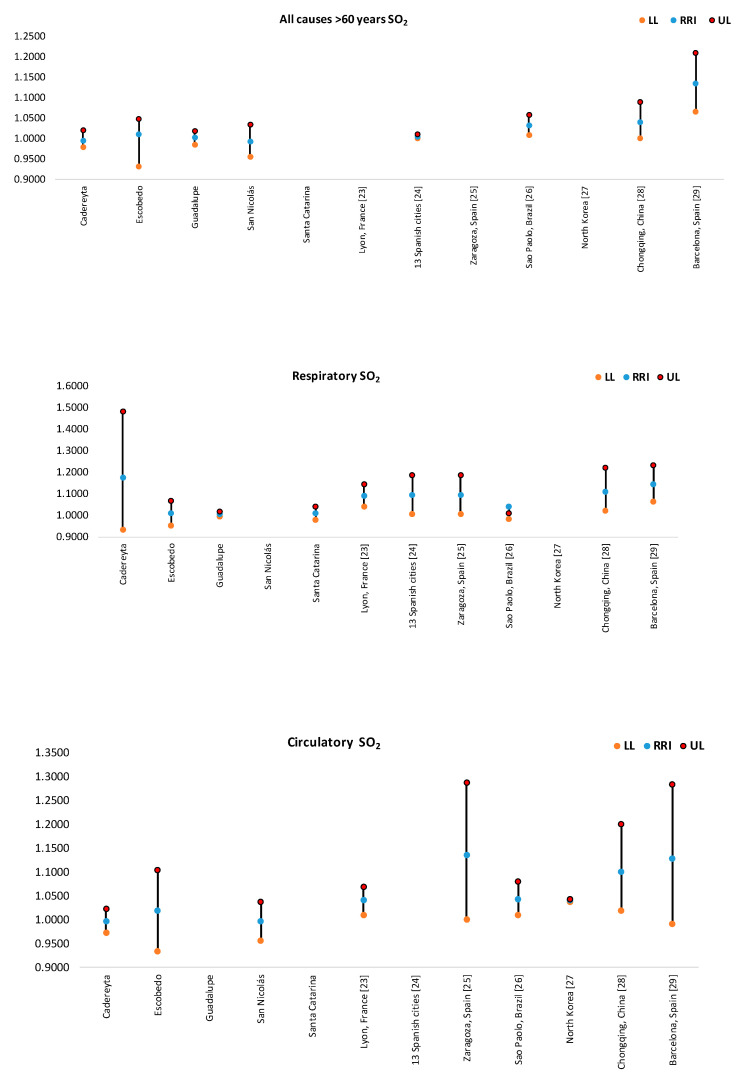
Comparison of RRI for daily mortality by an increase of 10% in SO_2_ concentrations. Not all cities had all the RRI values for each subplot.

**Figure 10 ijerph-17-09219-f010:**
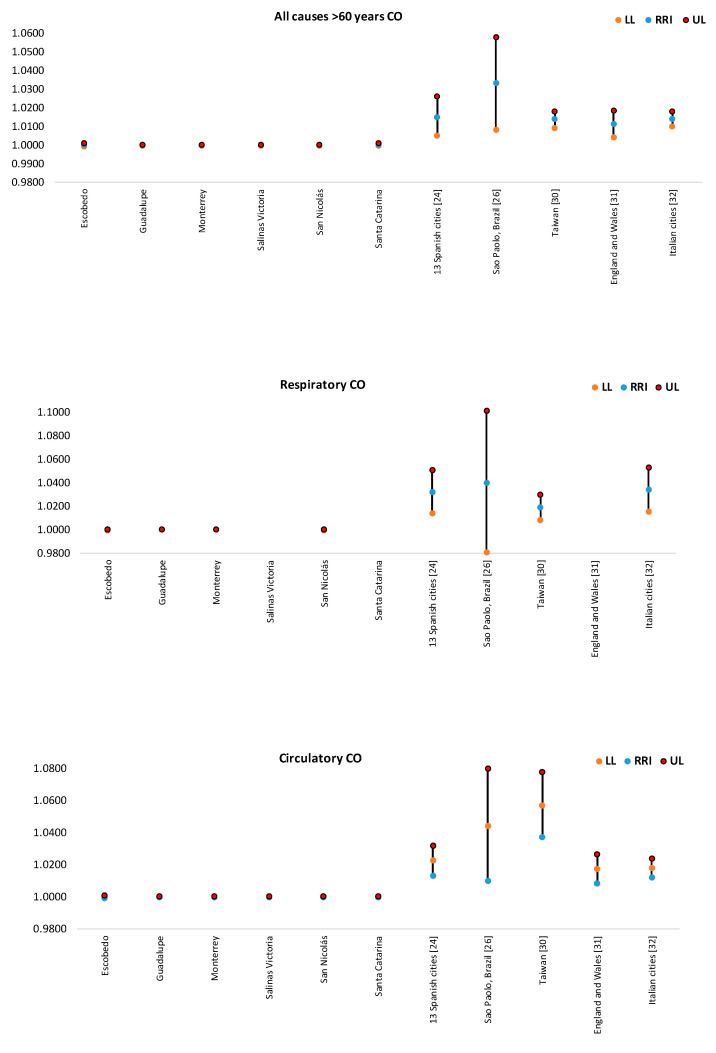
Comparison of RRI for daily mortality by an increase of 10% in CO concentrations found in this study with those reported by other authors. Not all cities had all the RRI values for each subplot.

**Figure 11 ijerph-17-09219-f011:**
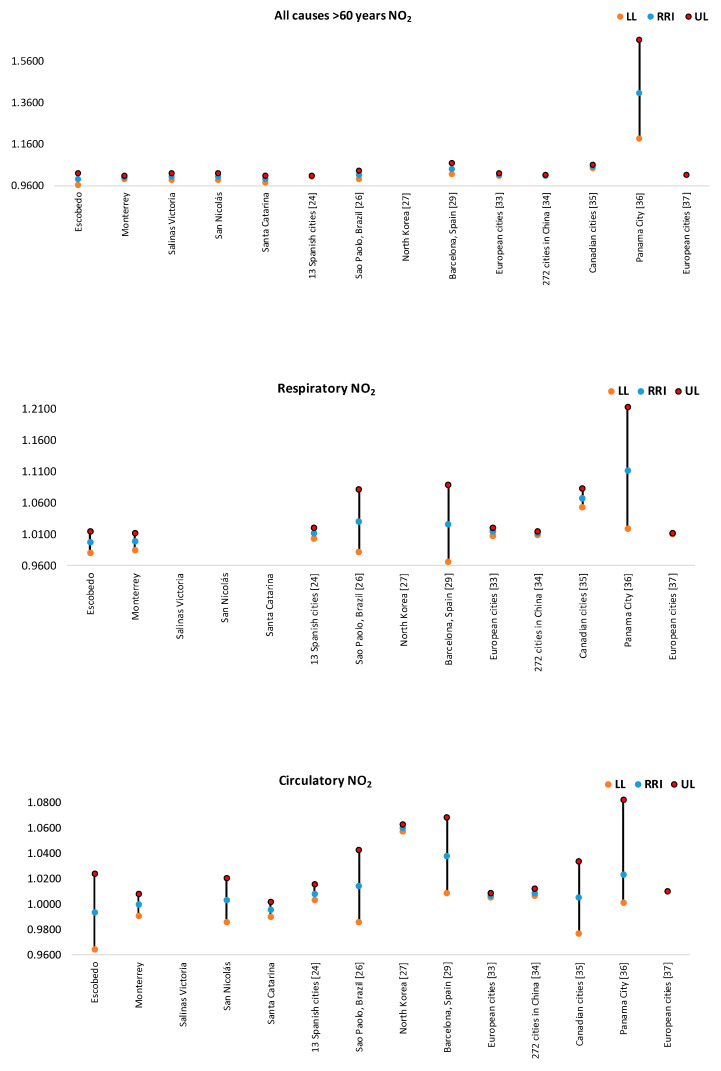
Comparison of RRI for daily mortality by an increase of 10% in NO_2_ concentrations found in this study with those reported by other authors. Not all cities had all the RRI values for each subplot.

**Figure 12 ijerph-17-09219-f012:**
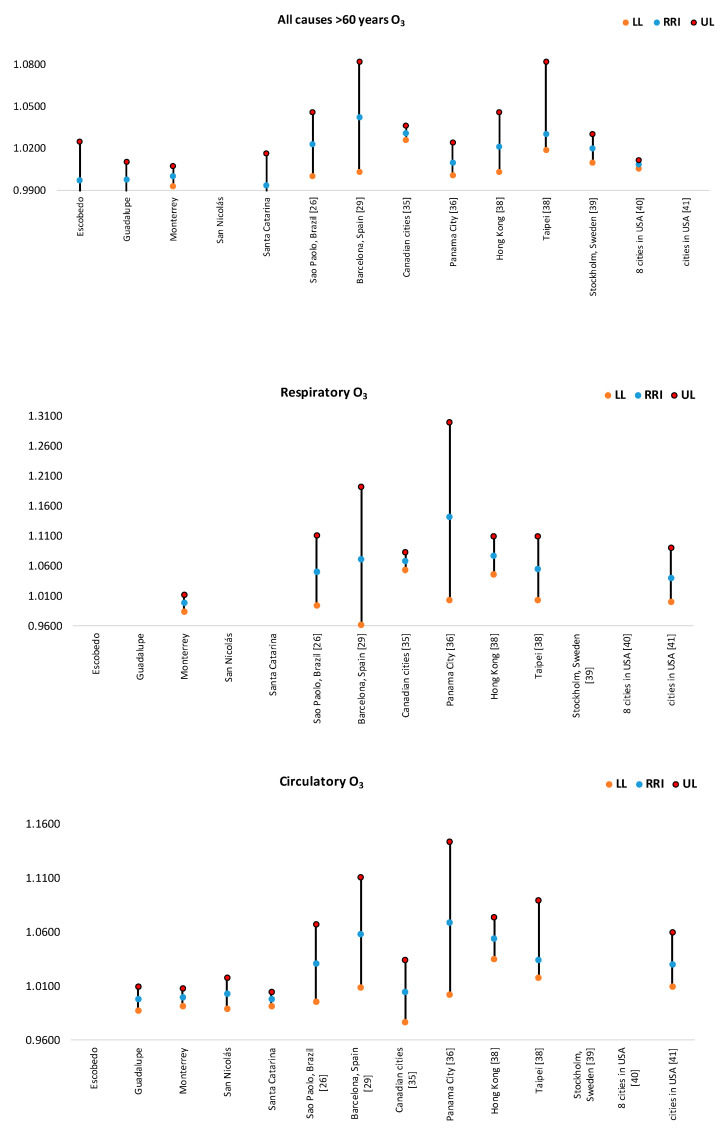
Comparison of RRI for daily mortality by an increase of 10% in O_3_ concentrations. Not all cities had all the RRI values for each subplot.

**Figure 13 ijerph-17-09219-f013:**
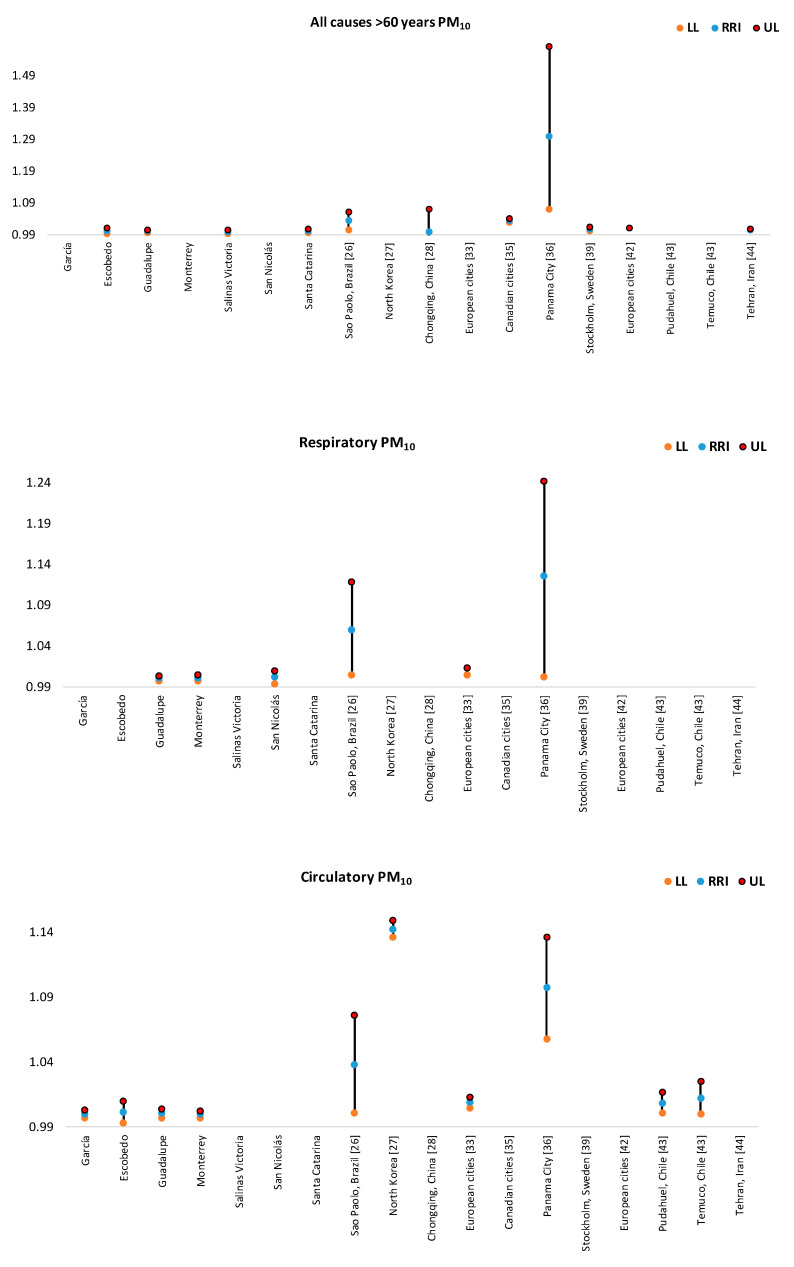
Comparison of RRI for daily mortality by an increase of 10% in PM_10_ concentrations found in this study with those reported by other authors. Not all cities had all the RRI values for each subplot.

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
