# Peer review of "Short-Term Effects of Atmospheric Pollution on Daily Mortality and Their Modification by Increased Temperatures Associated with a Climatic Change Scenario in Northern Mexico"

_ijerph, 2020, doi:10.3390/ijerph17249219_

Round 1

Reviewer 1 Report

The paper by Bretón et al. “Short-term effects of atmospheric pollution on daily mortality and their modification by increased temperatures associated with a climatic change scenario in Northern Mexico” used Poisson regression and multilinear regression to study the effects of air pollutant concentrations on daily mortality. The authors then compared the relative risks to similar studies done in cities around the world.

Figure 1
Scale bar needed. The fonts are too small to see.

L89
A website address needed here for the data source.

L102
Details of the imputations need to be included here or in supplemental materials.

Section 2.3
How are outliers/error values determined and removed? Friedman test only?

L119
The area size of each municipality should be included in the paper.

L127-130
Please include a formula here.

L131
Might be better to use “base model”.

L136 "up to 7 days"
Are all met variables adjusted by the same 7 days lag? Or only some? Please explain. And is the lag adjustment done for air pollutant predictors as well?

L168 “highest maximum value”
Is outlier removal done first?

L167-175
Please use less significant digits for the concentrations.

Figure 3
Fonts too small to see.

L192
What’s the sum of squares type III?

L194
What’s the meaning of F here?

Figure 4
Same problem as Fig. 3. The fonts are too small. Also, in the caption, should be 10% in mean daily concentrations.

Figures 5-9
Please enlarge the plot.
Not every city has an RRI value in each subplot. This should be noted in the caption.

Figure 6 is really surprising because every RRI is really small compared to other cities. CO is a tracer for incomplete combustion and can come from vehicles on the road. The cities in Mexico seem not that different from the referenced cities. Please conduct an additional literature search to find studies conducted in similar cities and compare the CO RRI again.

L317
I do not see where Figure 10 is. Please check.

Reviewer 2 Report

The manuscript is well-organised. However, some weaknesses make the paper not acceptable in its current form.
- In the introduction, it is suggested to introduce the the current situation of air pollution in the study Area.
- In Figure 1, the scale is missing.
- In  section 2.3,   what is the selection basis of time period "January 1 of 2012 to December 31 of 2015"? Is there any latest data?
- I suggest to introduce the notation of some acronyms, such as MCMC and  NIPALS.
- In section 2.5, the methods can be described in more detail.
- Figures 3 and 4 are not clear.
"All municipalities in MAM exceeded the reference value in the Air Quality Mexican Standards for O3 and PM10 (Figure 3)"   I didn't see the "O3 and PM10"  in Figure 3.
- In lines 179-180, what is "Table 3. 2. Association between Atmospheric Pollutants and Daily Mortality".
- In lines 293-295, what is "Figure 95".

Reviewer 3 Report

In manuscript IJERPH-1014120 short-term effects of air pollution on the health of residents in the Metropolitan Area of Monterrey (Mexico) were assessed from 2012-2015 using a time-series approach. Results outlined that guadalupe had the highest mean concentrations for SO2, CO and O3; whereas Santa Catarina showed the highest NO2 concentrations. Escobedo and Garcia registered the highest levels for PM10. Only PM10 and O3 exceeded the maximum permissible values established in the Mexican official standards. Most of pollutants and municipalities showed a great number of associations between an increase of 10% in their current concentrations and mortality, especially for people >60 years. Different scenarios resulting from climatic change were built (increases of 5-25% in daily mean temperature), but only the increase of 25% (5°C) showed a significant association with air pollutant concentrations and mortality. All pollutants and municipalities showed significant increases in relative risk indexes (RRI) resulting from an increase of 5°C when people >60 years was considered. Results were comparable to those reported by other authors around the world. The RRI were low but significant, and thus are of public concern. This study demonstrated that the elderly is strongly threatened not only by atmospheric pollution but also by climatic change scenarios in warm and semiarid places.

Overall, the manuscript is written in a clear exhaustive manner and with scientific soundness. The study, the objectives, and the obtained results are very interesting and represent an advancement of the current state of knowledge. The introduction provides sufficient elements to characterize the background and the problem statement. The study design, review methodology and the adopted inclusion/exclusion criteria are adequate for the purpose and clearly presented. The results obtained were presented in a clear and complete manner. The limits of the study are also presented and discussed clearly. Plese consider the following minor revisions:

  • Line 76 “According to Köppen and Geiger” Please add proper reference
  • The figures are of poor quality and the texts they contain practically illegible. Please consider improving the quality of the figures
